# Prediction of Shoreline Change for the Calculation of the Integrated Littoral Sediment Budget

**Yeon-Joong Kim**  **and Jong-Sung Yoon** *

Department of Civil and Urban Engineering, Inje University, Gimhae 50834, Korea; anyseason@inje.ac.kr
* Correspondence: civyunjs@inje.ac.kr; Tel.: +82-55-320-3434

**Abstract:** The severe coastal erosions are being accelerated along the east coast of South Korea owing to the intermittent erosions and depositions caused by the imbalance between the effective sediment volume supplied from coasts and rivers and the sediment transport rate. Consequently, many studies are being conducted to develop coastal-erosion reduction measures. To accurately determine the cause of coastal erosion, the causes of the erosion and deposition should be accurately diagnosed, and a comprehensive evaluation system for the sediment transport mechanism in the watershed and sea while considering regional characteristics is required. In particular, realizing the evaluation of the effective sediment volume that flows from the river to the sea through observations is a highly challenging task, and various research and developments are required to realize it, as it is still in the basic research stage. The purpose of this study was to systematically analyze the comprehensive sediment budget for coastal areas. First, an analytical system was developed. Then, a shoreline model was constructed by considering the size of the mixed particles. The parameters required for developing the model were determined using the observation data to improve the shoreline model. A sediment runoff model was applied to evaluate the effective sediment volume supplied from the river to the sea, and the applicability of this model was evaluated by comparing it with the sediment supply volume according to the soil and water assessment tool model. The representative wave and the input parameters of the model were set using the observation data of several years. It was found that the prediction performance of the shoreline change model improved when the effective sediment volume was considered, and the particles of the sediment on the shore were assumed to comprise multiple sizes. In particular, the prediction performance improved when the balance of the sediment budget was adjusted by applying a groin having a structurally similar performance to take into consideration the geographic features of the Deokbongsan (island) in front of the river mouth bar. The model demonstrated a good performance in reproducing long-term shoreline changes when the characteristics of the sea waves and the effective sediment volume were considered.

**Keywords:** beach morphodynamics; shoreline change; effective sediment volume; one-line model

## 1. Introduction

The severe coastal erosions are being accelerated along the east coast of South Korea owing to the intermittent erosions and depositions caused by the imbalance between the effective sediment volume supplied from coasts and rivers and the sediment transport rate. The causes of coastal erosion can be primarily classified into natural coastal erosion, which occurs at a low intensity over a long period of time, and anthropogenic coastal erosion, which occurs suddenly owing to the construction of structures and development projects in coastal areas. The majority of maritime states in the world are conducting research to reduce coastal erosion due to various anthropogenic causes, such as the reduction of sediment supply from the land due to surface covering following industrialization and urbanization, river maintenance, dam and weir construction, disturbance of the longshore sediment transport system owing to the increase in coastal structures such as ports and revetments, and coastal sand collection, in addition to natural causes such as climate change, which

causes the cycle of surface erosion, sea-level rise, and increased frequency and intensity of storm surges.

The hydrodynamic processes of coastal waters, such as storm surges, tides, sea currents, and rising sea levels, act as external forces that cause short-term (e.g., seasonal to multiannual) morphological changes in beaches [1–4]. In contrast, long-term (e.g., decadal to centennial) shoreline changes are caused by natural (effective sediment volume) and artificial sediment supply (beach nourishment), sea-level changes, land use, and climate change [5–7]. Thus, mid- to long-term changes in the shoreline are caused by various factors. Therefore, in order to improve the ability to predict shoreline changes and the analysis of the corresponding mechanism, a monitoring analysis is crucial for identifying factors that cause shoreline changes at the scale of the highest possible frequency and longest possible time in a transport environment of a wide range of sediments that exhibit natural variability [8–10].

In the sea area where the beach is maintained, sediment transportation is caused by two forms of movements: longshore sand transport, which involves transportation parallel to the coastline along the left and right shorelines, and cross-shore sand transport, which involves transportation in the direction perpendicular to the coastline. Furthermore, in the sea area adjacent to a river, direct sediment supply (effective sediment discharge) occurs from the river, thus acting as a crucial parameter in determining the characteristics of the sediment budget (Figure 1). Beach deformation is particularly active in swash zones owing to the effects of waves; hence, sediment transport in swash zones is also being actively researched. The process of sediment flow from a river to a sea area has been researched (Milliman and Meade, 1983) [11], and an analysis of the littoral sediment budget according to wave changes was conducted using the Delft3D model (Kim et al. 2019) [12] and a sensitivity analysis of the parameters related to sediment transport (Yang and Son, 2019) [13]. Furthermore, research was conducted for predicting shoreline changes due to sediment transport, including an experimental study on shoreline changes in a beach process (Son and Lee, 2000) [14]; prediction of shoreline transformation through a numerical model (Park et al., 1993) [15]; erosion-control-line setting through HaeSaBeenN, a prediction model for shoreline changes due to high waves (Park et al. 2019) [16]; and prediction of shoreline change according to a sea-level-rise scenario due to climate change (Vitousek et al., 2017) [17]. Meanwhile, in the swash zone of the waves in the breaking-wave zone and outside the breaking-wave zone, sediments are transported as a bed load by the fluid motion due to waves. The need for research to clearly understand the hydraulic phenomenon in the swash zone, where significant transport of beach sediments occurs, is emphasized in order to reduce and manage disasters that frequently occur along the coast owing to extreme storms, super typhoons, and abnormal swells, as well as sea-level rise caused by climate change (Lee et al., 2019) [18]. Understanding and predicting shoreline changes from a comprehensive perspective is a critical task for shore managers and policy makers (Stive et al., 2002) [19].

Thus, an investigation of the causes of coastal erosion requires a comprehensive evaluation system for the sediment transport mechanism in watersheds and sea areas while taking into consideration the regional characteristics, as well as an accurate diagnosis of the erosion and deposition. Furthermore, although various shoreline-change prediction models and numerous observation data are used to establish coastal-erosion reduction measures, there still is a need for research on analysis systems for sediment budget—which is a quantitative evaluation index—the utilization of observation data, methods for taking into consideration regional characteristics, and the performance improvement of the shoreline model.

Therefore, the aim of this study is to improve the performance of the shoreline model by establishing an analysis system for the systematic analysis of the integrated sediment budget in a coastal zone, building a multi-particle-size shoreline model, and setting parameters required for the model using observation data. A sediment runoff model was applied to evaluate the effective sediment volume supplied from the river to the sea area, and

the applicability of the model was evaluated via a comparison with the sediment supply volume according to the soil and water assessment tool (SWAT) model. The representative wave and input parameters of the model were set using observation data collected over several years. The prediction performance of the shoreline change model improved when the effective sediment volume was considered, and multiple particle sizes were assumed for the sediments of the coast. In particular, the prediction performance improved when the balance of the sediment budget was adjusted by applying a groin that had a structurally similar performance in order to consider the topographical features of the Deokbongsan (island) in front of a river mouth bar. The long-term shoreline changes were reproduced well as a result of considering the wave characteristics and effective sediment volume acting on the sea area.

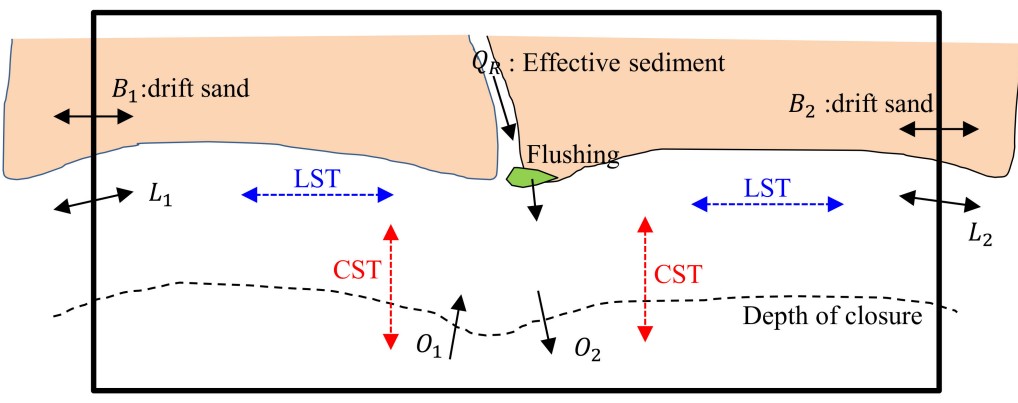

CST: Cross-shore sediment transport ($O_1$, $O_2$)  $B_1$, $B_2$ : drift sand
LST: Longshore sediment transport ($L_1$, $L_2$)  $Q_R$ : effective sediment from river

**Figure 1.** Principal components involved in littoral sediment budget.

## 2. Integrated-Sediment-Budget Calculation System

Rivers generally transport various materials such as surface water and sediments generated in watershed space from mountainous areas to river-mouth and sea areas, and shorelines change owing to the sediments flowing in from the river and those moving into the sea area owing to wave action. Sediments are supplied to the coast via a continuous flow from the rivers in mountainous areas; in particular, the majority of sediments that flow into rivers and to the sea area through rivers are generated by erosion resulting from intensive rainfalls and large-scale landslides (development of debris flow). In sea areas comprising river mouth bars, the closed sand bars collapse owing to the flow of the river during a flood, which causes a sudden discharge of sediments, and the collapsed sand bars gradually recover owing to the normal wave action; these collapses and recoveries occur repeatedly (Figure 2). Thus, if an anthropogenic or natural change occurs in the continuous flow field "upstream (mountain) ↔ mid-stream (river) ↔ downstream (coast)," environmental changes occur in the respective areas accordingly. However, until date, for sediment management, various measures have been established and managed for each area, including mountains, dams, rivers, and coasts. As a result, the effects of various measures for disaster mitigation and sediment management in each area are apparent, but many problems such as coastal erosion still occur along the coast. Therefore, it is crucial to first accurately analyze and calculate the sediment budget, which quantitatively represents the relationship between the inflow and outflow of sediments. However, because the interannual evaluation of the effective sediment volume supplied from the mountain and river areas is complex, empirical relations based on long-term observations and field surveys are mainly used, and the sediment yield is evaluated using an analysis model with a geographic information system (GIS). It is also important to evaluate sudden sediment runoff following the collapse of sand bars due to a flood wave and the sediment yield that is restored by an ordinary wave. Thus, the sediment-budget analysis method differs

according to the environment of the sea area. In particular, the integrated sediment budget is required to be analyzed (Figure 3) in a sea area where sediments are directly supplied from a river and there is a river mouth bar.

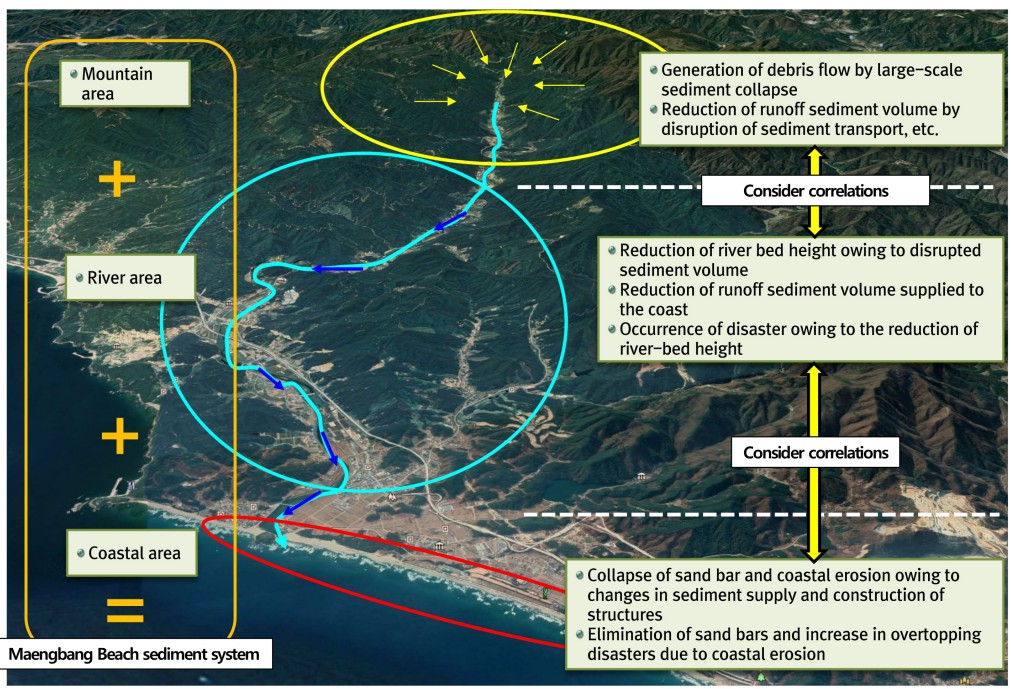

**Figure 2.** Conceptual diagram of integrated sediment management (example: Maengbang Beach).

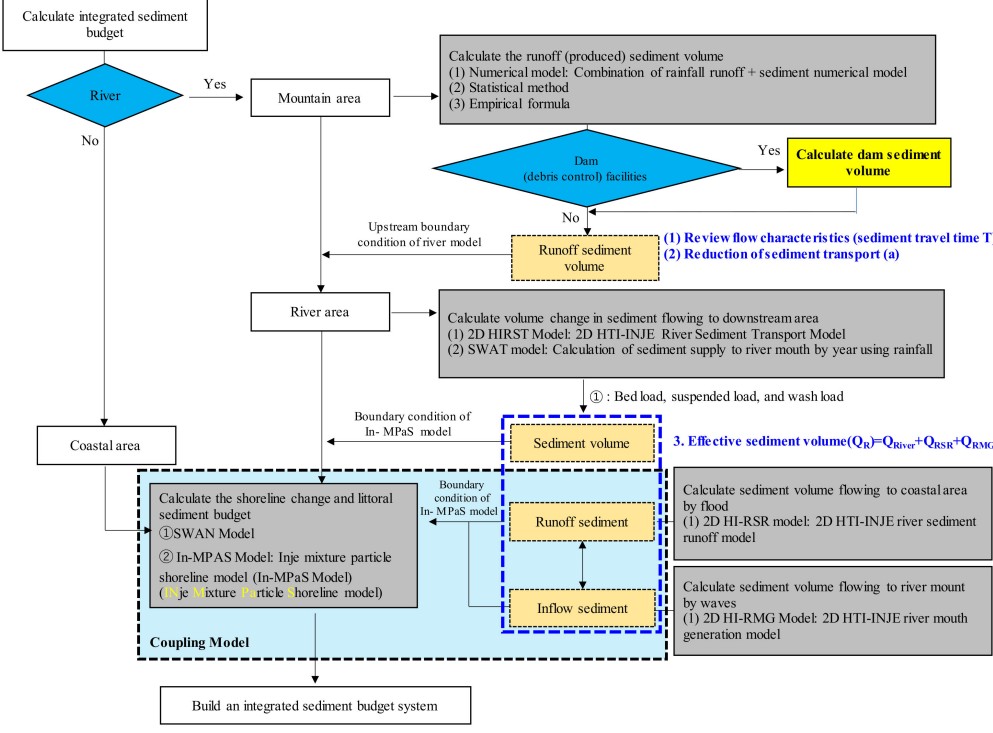

**Figure 3.** Flowchart of the integrated sediment budget system.

To evaluate the interannual effective sediment volume supplied to the sea area, in this study, the sediment volume was comparatively evaluated using the sediment flow calculation formula based on parameters indicating regional characteristics and topographic

information and using a numerical analysis based on GIS information (SWAT model). The results thus obtained were used as the input data for the shoreline model.

*Study Site*

Maengbang Beach, the study site, is one of the areas that are in need of measures for reducing coastal erosion. The coastal erosion condition of the beach was assessed as "serious" in 2016 based on the data of the Ministry of Oceans and Fisheries (2016) [20]. Located in Samcheok City in Gangwon-do, Maengbang Beach comprises a quasi-straight open coast with an average slope of approximately 45° from the north. Maengbang Beach is to the north of Samcheok city, and Deoksan Beach is to the south. Sediments are supplied to the sea area from Deokbongsan and Maeupcheon, which are between the two beaches (Figure 4). Maeupcheon has a catchment area of 143.83 km$^2$, a channel length of 30.98 km, an average catchment elevation of 414.21 m, and an average catchment slope of 28.14°.

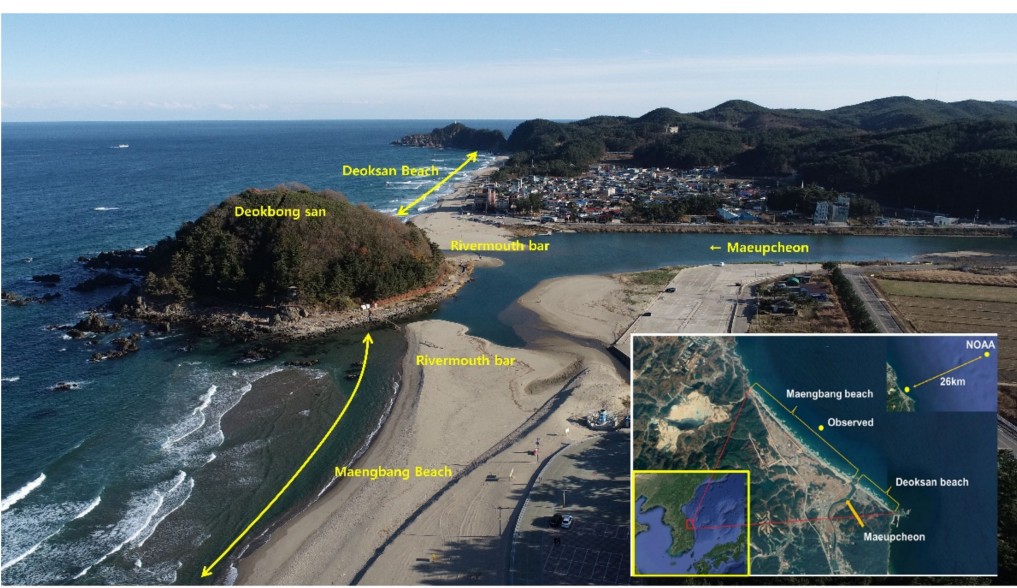

**Figure 4.** Study site (Maengbang Beach, Gangwon-do, Korea).

To investigate the characteristics of the river mouth bar, which changes with the river flow during a flood, continuous observation was performed with closed-circuit television (CCTV) cameras (Figure 2). In September 2020, heavy rain was caused by Typhoon No. 9, "Maisak". At this time, an overflow to Deoksan Beach was observed via the CCTV cameras. The sediments from Maeupcheon (effective sediment volume) flowed toward Maengbang Beach, and the direct sediment supply to Deoksan Beach was blocked by topographical features. However, Kim et al. (2020) found that an overflow to Deoksan Beach occurred when the flow rate of Maeupcheon during a flood exceeded approximately 600 m$^3$/s, and when it exceeded approximately 1050 m$^3$/s, the river mouth bar between Maengbang Beach and Deokbongsan collapsed, causing a sudden sediment runoff.

The wave data of the National Oceanic and Atmospheric Administration (NOAA) WaveWatchIII (2019) [21] model were collected to analyze the wave characteristics of Maengbang Beach. The wave data, collected from 1979 till date, comprise the wave height, period, wave direction, etc., at 3 h intervals in 0.5° resolution. The external force that generated and developed the river mouth bar was calculated using the wave data collected for approximately 40 years from the NOAA WaveWatchIII grid (longitude: 129.5°, latitude: 37.5°), which is located approximately 26 km offshore and is close to Maengbang Beach. Images obtained from the CCTV camera and field observations in Figure 5 show the characteristics of sudden shoreline change due to natural causes around the river mouth bar of Maengbang Beach during a flood.

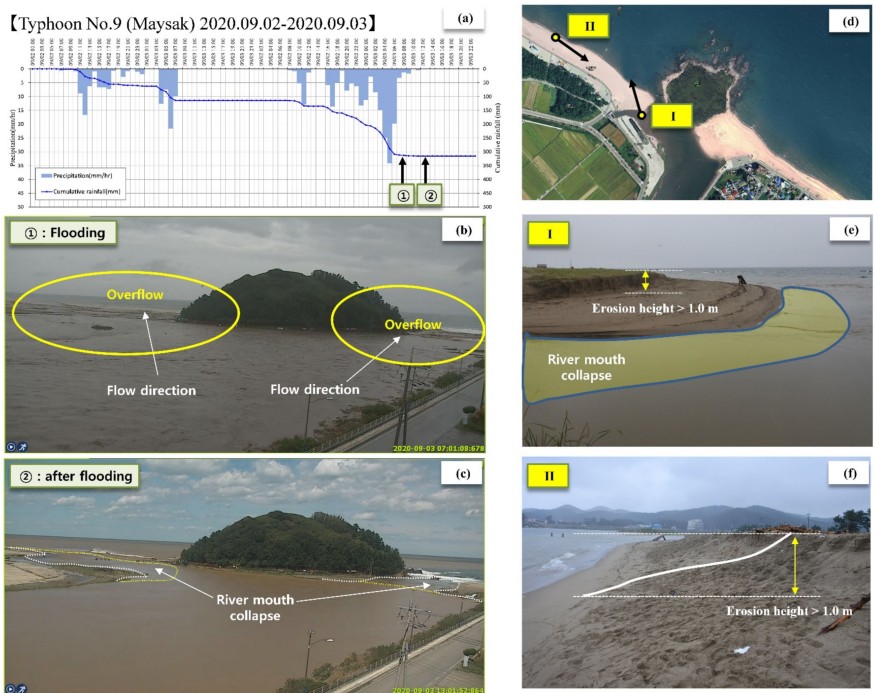

**Figure 5.** Characteristics of tidal sand bar in Gangwon-do Maengbang Beach: (**a**) cumulative rainfall (typhoon No. 9 Maisak in 2020), (**b**) overflow characteristics near river mouth (point ①), (**c**) changes in river mouth topography after flood (point ②), (**d**) location map at time of photographing, (**e**) occurrence of coastal erosion after flood (direction I), and (**f**) occurrence of coastal erosion after flood (direction II).

## 3. Shoreline Change Model Considering Multiple Particle Sizes

The aim of this study is to predict shoreline changes by calculating the flow characteristics that change in the wave-current co-existing fields (mountain, river, and coast areas) and the sediment yield discharged into the sea through rivers (effective sediment volume) and then analyzing the wave and topographic characteristics of sea areas. To predict the shoreline change of Maengbang Beach in Gangwon-do, the input parameters of the model were applied by carefully analyzing long-term observation data.

### 3.1. Effective Sediment Volume

The sediment runoff volume at a specific point in a river water system can be defined as the sediment runoff volume that passes through that point. It is highly challenging to accurately identify the sediment yield transported by surface water in a real catchment, and it is almost impossible to collect reliable data. Therefore, generally, a relation is determined between the water level and flow rate, which are relatively easy to observe, and it is easy to evaluate the moving sediment volume using a flow–sediment curve while considering the proportion (concentration) of sediments. In this case, the effective sediment volume is evaluated by applying various numerical analysis models. The evaluation methods for the effective sediment volume are described in the following sections.

3.1.1. Sediment-Runoff Empirical Formula

Specific sediment yield (specific production sediment yield and runoff sediment yield/catchment area) is a concept used for understanding the average erosion strength of an entire catchment. It started with a study that directly compared the average erosion strength value with a river catchment and considered it as the sediment runoff characteristic of each river catchment. In Japan, the following empirical formula was proposed based on long-term observation data (Ashida and Okumura, 1974) [22]. Although the prediction results differ depending on the accumulated data, and there are some difficulties in realizing

reliability and calculating the sediment runoff intensity coefficient by region, it is easier to approximately identify the annual runoff sediment yield (Figure 6). This empirical formula can be expressed as follows:

$$q_s = KA^{-0.7} \tag{1}$$

where $q_s$ is the annual mean specific sediment yield $(\text{m}^3/\text{km}^2/\text{year})$, $A$ is the catchment area $(\text{km}^2)$, and $K$ is the sediment runoff intensity coefficient.

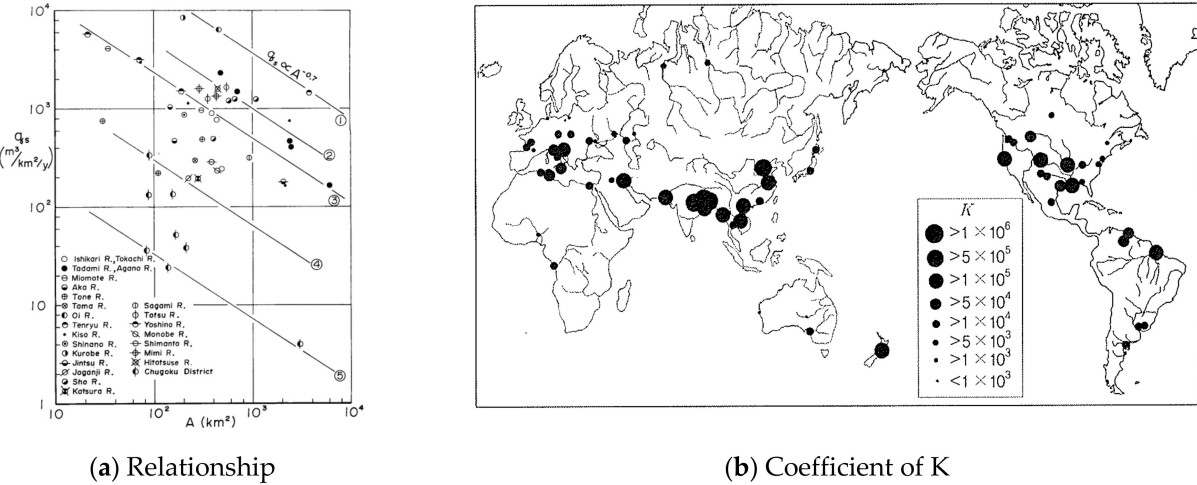

(**a**) Relationship

(**b**) Coefficient of K

**Figure 6.** Relation between specific sediment yield and catchment area in Japan.

### 3.1.2. Sediment Runoff Model

A sediment runoff model is constructed for long-term prediction through a quantitative analysis of the causal relationship between the sediment transport in a catchment and the river bed height. To assess the runoff sediment yield from mountain and river areas, the interannual effective sediment volumes are evaluated by calculating various parameters for the catchment characteristics of the study site using a sediment runoff model (Kato et al. 2018) [23] according to the geographic characteristics of rainfall and catchment as follows:

$$V = K_m\left(A^{3/10}(\sin\theta_1)^{9/20}(\sin\theta_2)^{3/10}R_e{}^{9/5}\right) \tag{2}$$

$$K_m = 0.007\left(A_r\left(\frac{\rho w}{\sigma}\right)\right)^{0.64} \tag{3}$$

where $V$ is the runoff sediment yield $(\text{m}^3)$, $K_m (= 10^{0.6})$ is the sediment runoff coefficient $(10^0 \sim 10^2)$, $A$ is the catchment area $(\text{km}^2)$, $\theta_1$ is the mean catchment slope, $\theta_2$ is the river slope, $R_e$ is the effective rainfall, $\rho$ is the unit volume weight of the soil particles $(2.65\,\text{t/m}^3)$, $\sigma$ is the compressive strength $(\text{t/m}^2)$, $w$ is the porosity, and $A_r$ is the river area $(\text{m}^2)$. Figure 7 presents the calculation result of the effective sediment volume for the mean rainfall $(1316\,\text{mm/y})$ in the Maeupcheon catchment. Here, $CN$ denotes the dimensionless runoff curve number, and $S$ denotes the potential maximum retention.

### 3.1.3. SWAT Model

The SWAT model is a catchment unit model developed by the Agricultural Research Service (ARS) of the United States Department of Agriculture (USDA). This model was created by combining ARS models that have been developed by the USDA, such as the CREAMS, GREAMS, and EPIC models. Weather data that change over time, such as daily precipitation, temperature, wind speed, sunshine, and relative humidity, and spatially changing data, such as land use status, soil properties, and topographic data, are required to spatiotemporally analyze the hydrology and water quality using the SWAT model [24–26]. As a continuous semi-distributed model, the SWAT model basically l includes four submodels: hydrology, soil loss, nutrients, and channel tracking. The input data

of the SWAT model (Figure 8) include topographic data(Figure 8A), soil data(Figure 8B), and land use data(Figure 8C), which are automatically configured using GIS and spatial information; weather, channel tracking, agricultural management, and groundwater data, which are manually input; and subwatershed data generated using GIS and manual input. In particular, the SWAT model can predict and simulate the rainfall runoff, movement of sediments, and other processes. It can also perform simulations even in unmeasured areas and can quantify the relative effect of water quality according to changes in the cultivation type, climate, and vegetation (Smithers and Engel, 1996) [27].

| Basin | Geographic information | | | | | | Effective rainfall | Sediment coefficient | Runoff volume |
| | Area | Overall slope | River slope | River area | CN | S | (Re) | | V |
| | (km$^2$) | $\theta_1$(°) | $\theta_2$(°) | A$_r$(m$^2$) | | | mm/day | km | (m$^3$/year) |
|---|---|---|---|---|---|---|---|---|---|
| ① | 6.19 | 13.71 | 6.93 | 4200 | 67.45 | 122.58 | 3.232 | 0.376 | 543.8 |
| ② | 16.37 | 22.91 | 9.1 | 7210 | 67.92 | 120.00 | 3.239 | 0.532 | 1400.6 |
| ③ | 4.5 | 11.17 | 4.29 | 4110 | 68.38 | 117.45 | 3.246 | 0.371 | 388.7 |
| ④ | 7.82 | 12.92 | 8.28 | 26,100 | 68.85 | 114.94 | 3.253 | 1.212 | 1952.2 |
| ⑤ | 26.3 | 6.38 | 2.37 | 77,400 | 69.31 | 112.47 | 3.260 | 2.430 | 2839.1 |
| ⑥ | 19.65 | 14.93 | 2.49 | 66,600 | 70.66 | 105.47 | 3.280 | 2.207 | 3540.1 |
| ⑦ | 8.85 | 12.44 | 6.1 | 7020 | 72.01 | 98.73 | 3.300 | 0.523 | 805.2 |
| ⑧ | 7.9 | 7.44 | 2.26 | 9900 | 72.12 | 98.18 | 3.302 | 0.652 | 573.4 |
| ⑨ | 10.42 | 6.88 | 3.52 | 15,600 | 72.24 | 97.63 | 3.303 | 0.872 | 919.9 |
| ⑩ | 3.16 | 19.73 | 6.51 | 2280 | 72.35 | 97.08 | 3.305 | 0.255 | 360.2 |
| ⑪ | 2.6 | 7.48 | 0.92 | 9360 | 72.46 | 96.54 | 3.306 | 0.629 | 304.2 |
| Σ | | | | | | | | | 13,627 |

**Figure 7.** Evaluation of effective sediment volume using a sediment runoff model.

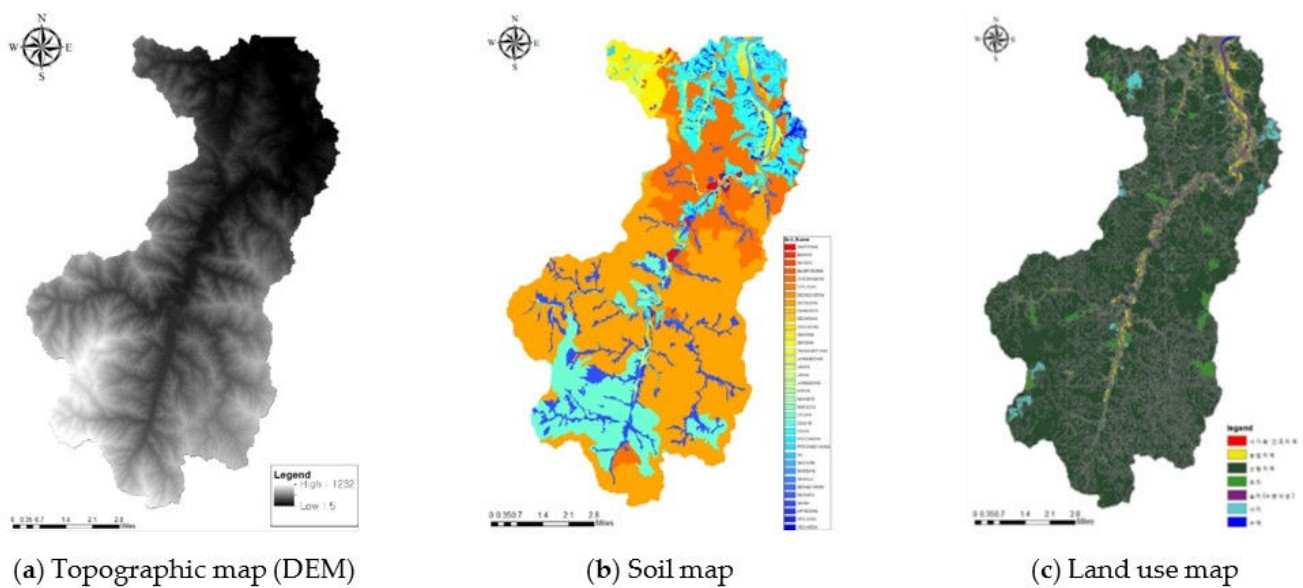

**(a) Topographic map (DEM)** **(b) Soil map** **(c) Land use map**

**Figure 8.** Main input data of the SWAT model (Maeupcheon catchment).

### 3.2. Building a Shoreline Model (IN-MPaS Model)

A shoreline model (In-MPaS, Inje University-Mixture Particle Shoreline model) was built, which could consider single-particle-size ($D_{50}$) and multi-particle-size distributions for sediments by using a numerical model for practical shoreline-change prediction based on the one-line theory (Hanson, 1989; Tomasicchio et al. 2020) [28,29].

### 3.2.1. Coefficient of Longshore Sediment Transport Rate by Particle Size

The longshore sediment transport rate by particle size is required to be calculated to consider the classification process of the multi-particle-size model. In addition to the existing CERC-type sediment transport equation, the relational Equation (4) suggested by Kamphuis et al. (1986) [30] for the sediment transport coefficient $K_1$ and low-quality particle size $D$ was used:

$$K_1 \propto = D^{-1/2} \tag{4}$$

$K_1$ indicates the degree of particle movement. If the particle size $(D)$ is large, the particle movement slows down; as a result, $K_1$ and the longshore sediment transport rate $Q$ decrease. Furthermore, if the shoreline is long, assuming that it is the same as the volumetric content $\mu^{(K)} = (K = 1 \sim N)$ by particle size (N), the flux of the wave energy acting on Equation (5) by particle size can be expressed via Equation (6) (Figure 9):

$$D^{(1)}, D^{(2)}, \cdots \cdots, D^{(n)} \quad (K = 1 \sim N) \tag{5}$$

$$F_x \mu^{(1)}, F_x \mu^{(2)}, \cdots \cdots, F_x \mu^{(N)} \quad (K = 1 \sim N) \tag{6}$$

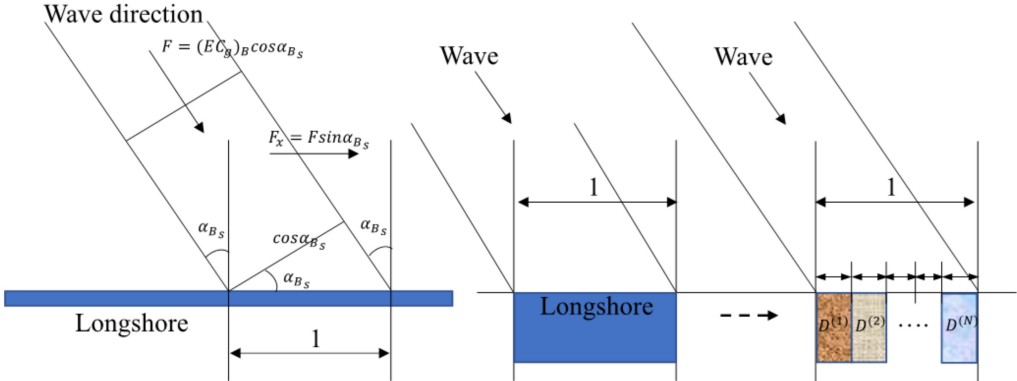

**Figure 9.** Wave energy flux acting on unit width of the shoreline.

Therefore, the longshore sediment transport rate by multi-particle size can be determined as follows:

$$Q^{(K)} = \mu^{(K)} K_1^{(K)} F_x \quad (K = 1 \sim N) \tag{7}$$

$$F_x = \left(EC_g\right)_B \sin \alpha_{Bs} \cos \alpha_{Bs} \tag{8}$$

$$K_1^{(K)} = A \left(D^{(K)}\right)^{-1/2} \quad (K = 1, 2, \cdots, N) \tag{9}$$

Here, $\left(EC_g\right)_B$ is the wave energy flux and coastal direction component at the wave breaking point per unit width of shoreline, and $\alpha_{Bs}$ is the angle formed by the breaking line and the shoreline at the time of wave breaking. The coefficient A in Equation (9) specifies the sediment transport coefficient, which is set in accordance with the transformation process of the target coast.

### 3.2.2. Sediment Volume Conservation Equation by Particle Size

The shoreline change $y_s^{(K)}$ by particle size is calculated by the following sediment volume conservation equation by particle size (continuous equation):

$$\frac{\partial y_s^{(K)}}{\partial t} = -\frac{1}{D_s} \left(\frac{\partial Q^{(K)}}{\partial x} - q^{(K)}\right), \ K = 1, 2, \cdots, N \tag{10}$$

where $x$ is the coordinate of the coast direction, $D_s$ is the movement height of the sediments, and $q^{(K)}$ is the sediment inflow (here, river runoff sediment yield) by particle size per unit

width of the shoreline, and the shoreline change $Y_s$ of the multi-particle size (full particle size) is calculated by the following equation:

$$\frac{\partial Y_s}{\partial t} = \sum_{K=1}^{N} \frac{\partial y_s^{(K)}}{\partial t}, \ K = 1, 2, \cdots, N \tag{11}$$

### 3.2.3. Change in Content by Particle Size

Sands are mixed by wave action, and changes in particle size occur in an exchange layer of a certain thickness. The change in the mixing rate of the sand particle size was calculated by Hirano (1971) [31] using the concept of the exchange layer for the erosion and deposition phenomena in coastal areas. The exchange layer was defined by the range of width $B$ in the coastal direction and mixing depth $D_s$ while assuming a beach section with a sediment moving height of $D_s$ and a constant beach slope of $tan\beta$, in accordance with the shoreline change model, as shown in Figure 10. The geometric relationship between the exchange layer width $B$ and mixing depth $\Delta D_s$ can be expressed as follows:

$$B = \Delta D_s \frac{1}{\tan \beta} = \Delta D_s \cot \beta \tag{12}$$

where the parameters of $B$ and $\Delta D_s$ were determined by referring to the study Kraus (1985) [32]. In the shoreline change model, the beach section moves parallel to the shoreline as the shoreline moves forward and backward. After a shoreline change, the sediments in the exchange layer $B$ are immediately mixed, but the mixing state appears to differ in the deposition and erosion areas. After the shoreline change, mixing occurs in the range of width $B$ toward the coast from the shoreline. Hence, in the deposition area (Figure 11), a new layer ① is deposited, and layers ① and ② are mixed, while the layer ③ of the coast side in $B$ is not mixed. In the deposition area, the mean particle size decreases as many sediments of small particle sizes are deposited based on Equations (7)–(10). Meanwhile, in the erosion area, layer ① is cut, and mixing occurs in layer ③ in the direction of the coast inside $B$, which is further inside than layer ② in the direction of the coast. At this time, many sediments of small particle sizes are released from the erosion area, and large particles remain in the exchange layer, thus increasing the overall mean particle size.

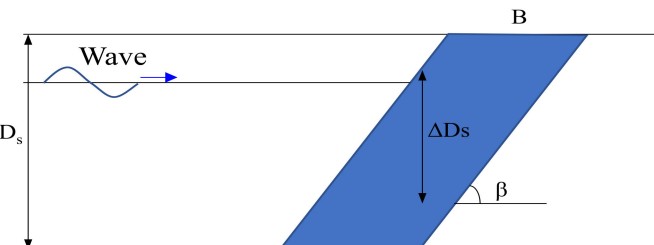

**Figure 10.** Definition of exchange layer mixed by wave action.

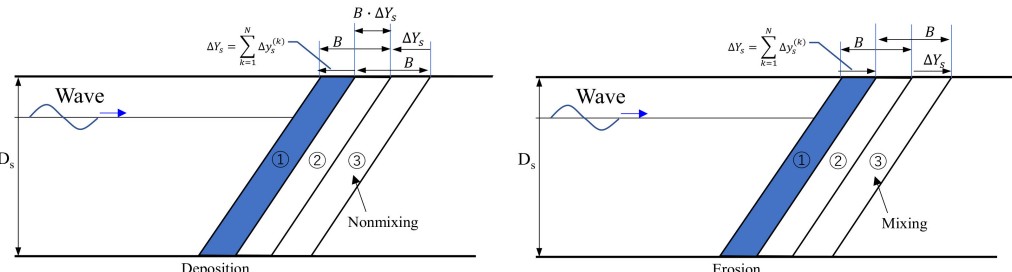

**Figure 11.** Particle-size mixing range after shoreline change (**left**: deposition, **right**: erosion).

The change in the content $\mu^{(K)}$ in the exchange layer by particle size was expressed in the form of basic equations by formulating the sediment budget by particle size in the layer based on the sediment inflow and outflow for each change in the erosion and deposition areas. These basic equations are as follows:

$$\text{Deposition}: \frac{\partial \mu^{(K)}}{\partial t} = \frac{1}{B} \left\{ \frac{\partial \mu_s^{(K)}}{\partial t} - \frac{\partial Y_s}{\partial t} \cdot \mu^{(K)} \right\} \tag{13}$$

$$\text{Erosion}: \frac{\partial \mu^{(K)}}{\partial t} = \frac{1}{B} \left\{ \frac{\partial \mu_s^{(K)}}{\partial t} - \frac{\partial Y_s}{\partial t} \cdot \mu_B^{(K)} \right\} \tag{14}$$

where $\mu^{(K)}$ is the content closer to the coastal side than the exchange layer before a shoreline change.

### 3.2.4. Numerical Calculation

The longshore sediment transport rate by particle size (Equation (4)) was calculated for the content distribution of the initial shoreline and initial particle size and for the incident wave condition, and the calculated longshore sediment transport rate by particle size was assigned to the left and right sides (shores) of the coast.

$$Q^{(K)} = Q_B^{(K)} \quad (K = 1, 2, \cdots, N) \tag{15}$$

However, because the particle size content changes over time at the boundary where the littoral sediments run off to the outside in the calculation process, the distribution of the longshore sediment transport rate by particle size changes over time. Although the overall longshore sediment transport rate may decrease, the longshore sediment transport rate by particle size cannot be reduced in advance. Therefore, in this calculation, the overall longshore sediment transport rate $(Q_B)_{all}$ at the boundary was assigned, and the longshore sediment transport rate by particle size was determined by calculating the distribution ratio using the content at the adjacent point of the boundary at each time according to the following equation:

$$Q^{(K)} = \frac{\mu^{(K)} K_1^{(K)}}{\sum_{K=1}^{N} \mu^{(K)} K_1^{(K)}} \cdot (Q_B)_{all} \quad (K = 1, 2, \cdots, N) \tag{16}$$

In the case of the sediment inflow (effective sediment volume) from the river mouth to the coast, the sediment inflow from the outside $q^{(K)}$ in the basic equations was applied, and the river inflow sediment volume by particle size $q_{in}^{(K)}$ was supplied to $q^{(K)}$ as follows:

$$q^{(K)} = q_{in}^{(K)} \quad (K = 1, 2, \cdots, N) \tag{17}$$

When no flow occurs because the littoral sediments are blocked by protruding structures such as groins installed on the shoreline, the longshore sediment transport rate at that point is set as zero. The equation for the change in content by particle size in the exchange layer is as follows. The sediment budget by particle size in the layer due to sediment inflow and outflow can be obtained by formulating it with a shoreline change at each time for the deposition and erosion areas, respectively. Figure 12 presents the calculated flow of the shoreline change prediction model while considering multiple particle sizes.

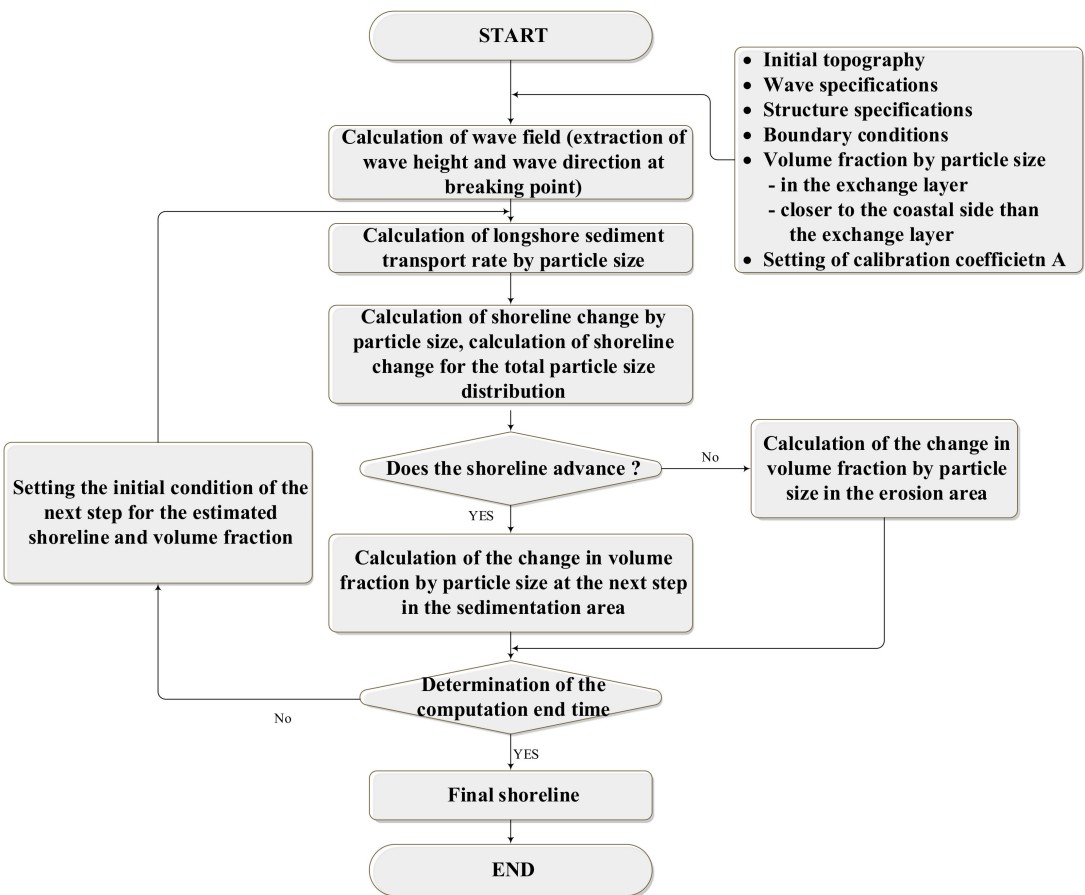

**Figure 12.** Calculation flow of the shoreline-change prediction model.

## 4. Methods

### 4.1. Depth Survey

The depth and beach section surveys were conducted via a survey ship equipped with a real-time-kinematic global navigation satellite system (RTK GNSS), high-precision GNSS, and a precision echo sounder (AquaRuler 200S, MIDAS) from MSL (−) of 25 m outside the breaking wave zone to the MSL (+) of 6 m, the reference point above sea level. On measuring the depth 22 times for approximately 34 months from March 2017 to December 2019, almost no topographical change was observed outside the MSL (−) of 10 m. Hence, it is believed that the depth of the closure was formed at an MSL (−) of approximately 10 m (Jin et al., 2020) [33]. The analysis result of the depth data indicates that the variability of seasonal wave characteristics and the topographical variability of intermittent high waves and arc-shaped sand bars are significant. Therefore, transverse and longshore sediments and sediment transport along the beach circulation should also be taken into consideration in order to accurately predict the behavior of the arc-shaped sand bars on Maengbang Beach and the resulting beach response. The sediment budget was analyzed by preparing interannual beach cross-sections for each lateral line using long-term depth survey results (Figure 13). This result was compared with the result of the sediment budget using the shoreline model to evaluate the reliability of the model.

### 4.2. Shoreline Analysis

The characteristics of interannual shoreline changes (Figure 14) were identified by comparing the satellite photographs of Maengbang Beach (1971–2017). To understand the forward and backward movements of the interannual shorelines at Maengbang Beach and Deoksan Beach, the road boundary in the latest satellite image obtained after the coastal maintenance was completed was set as the baseline for the shoreline change, and the

shoreline change was analyzed by applying it to past satellite images. In addition, as shown in Figure 13A, the position of the shoreline for each lateral line was determined by dividing the entire sea area into 225 offshore sections at 25 m intervals (total length of 5875 m). This result was used as the initial topographic condition (shoreline) and verification data of the shoreline change prediction model.

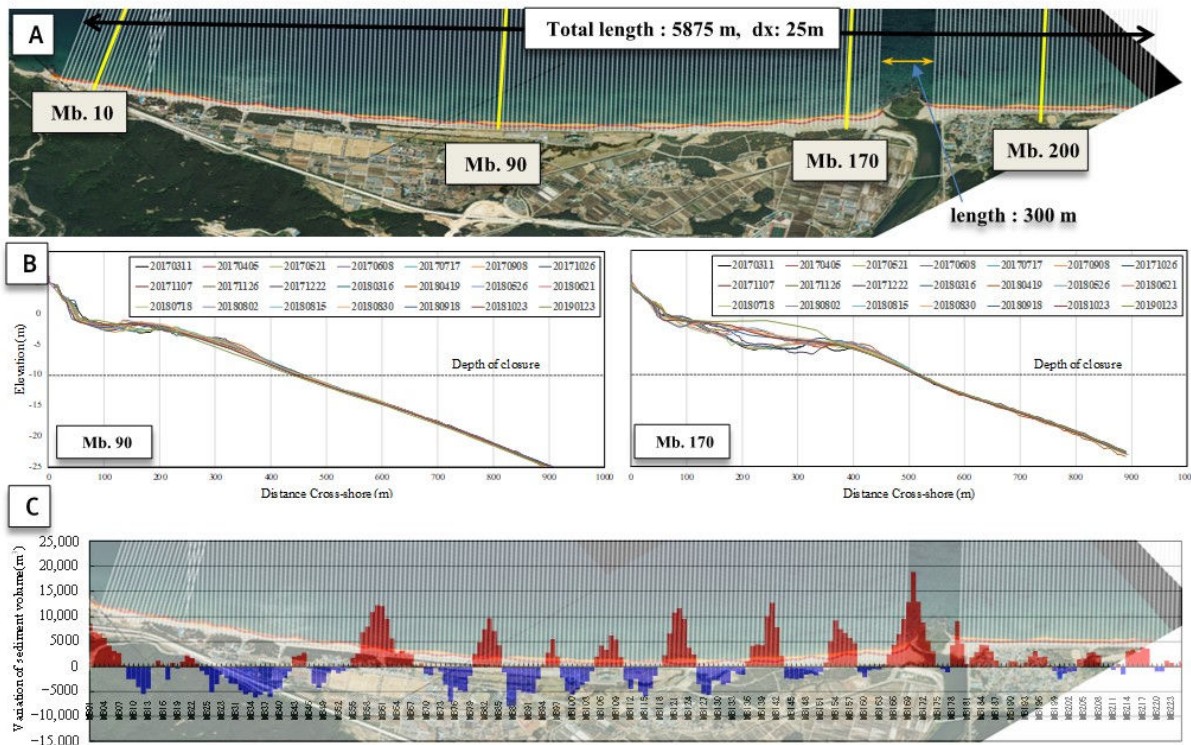

**Figure 13.** (**A**) Lateral lines of Maengbang Beach; (**B**) beach cross-sections according to depth survey result; (**C**) analysis result of sediment budget variations from March 2017 to January 2019 (red: deposition and blue: erosion).

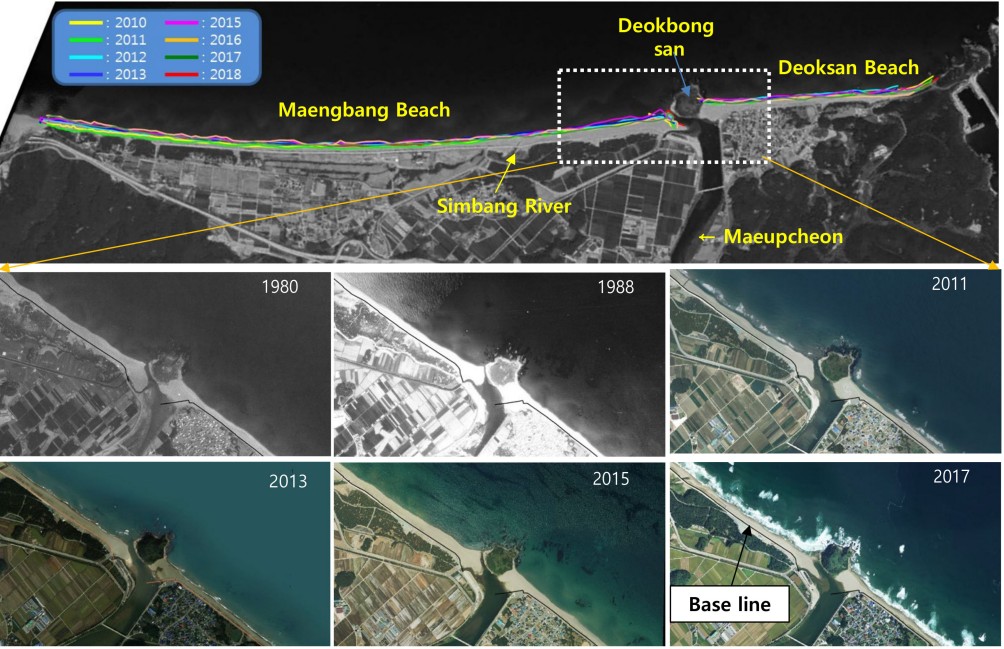

**Figure 14.** Shoreline change observed in satellite images.

### 4.3. Wave Characteristics

The wave data of the NOAA WaveWatch III (2019) model were collected to analyze the wave characteristics of Maengbang Beach. The wave data provide the wave height, period, wave direction, etc., at 3 h intervals at 0.5° resolution from 1979 until recently (2019). The characteristics of waves entering Maengbang Beach were analyzed using wave data collected by the NOAA for approximately 40 years (longitude: 129.5°, latitude: 37.5°), which is closest from Maengbang Beach (approximately 26 km offshore, Figure 4).

Shoreline transformation is mainly caused by intrusive waves from the open sea, which include many wave components with a high wave height and long period. The isometric changes for predicting long-term beach transformations are calculated using the energy-averaged waves and mean tidal level, which represent the annual wave energy. Toril (2003) [34] suggested the use of energy-averaged waves for the wave components to predict shoreline changes due to coastal structures. In this study, the representative wave of the sea area was applied by calculating the energy-averaged wave (Equation (18)) to predict the interannual shoreline change. The results are summarized in Table 1, and the energy-averaged wave-direction rose diagrams by season are presented in Figure 15:

$$H_m = \sqrt{\frac{\sum_{i=1}^{N}\left(H_i^2 \cdot T_i\right)}{\sum_{i=1}^{N} T_i}}, \quad T_m = \frac{\sum_{i=0}^{N}(T_i)}{N}, \quad \theta_m = \cos^{-1}\left(\frac{\sum_{i=0}^{N}\left(\cos\theta_i \cdot H_i^2 \cdot T_i\right)}{\sum_{i=1}^{N}(H_i^2 \cdot T_i)}\right) \quad (18)$$

where $H_i$, $T_i$, and $\theta_i$ denote the observed values of the wave height, period, and wave direction, respectively, and $H_m$, $T_m$, and $\theta_m$ denote energy-averaged wave height, period, and wave direction, respectively.

**Table 1.** Analysis results of wave characteristics (January 1979–May 2019).

| Track | Energy-Averaged Wave Height ($H_m$) | Period ($T_m$) | Wave Direction ($\theta_m$) | |
|---|---|---|---|---|
| Spring (Mar.~May.) | 1.104 | 5.702 | 89.998 | E |
| Summer (Jun.~Aug.) | 0.892 | 5.278 | 68.241 | ENE |
| Fall (Sep.~Nov.) | 1.356 | 6.271 | 43.906 | NE |
| Winter (Dec.~Feb.) | 1.572 | 7.039 | 4.075 | NE |
| All seasons | 1.283 | 6.069 | 45.326 | NE |

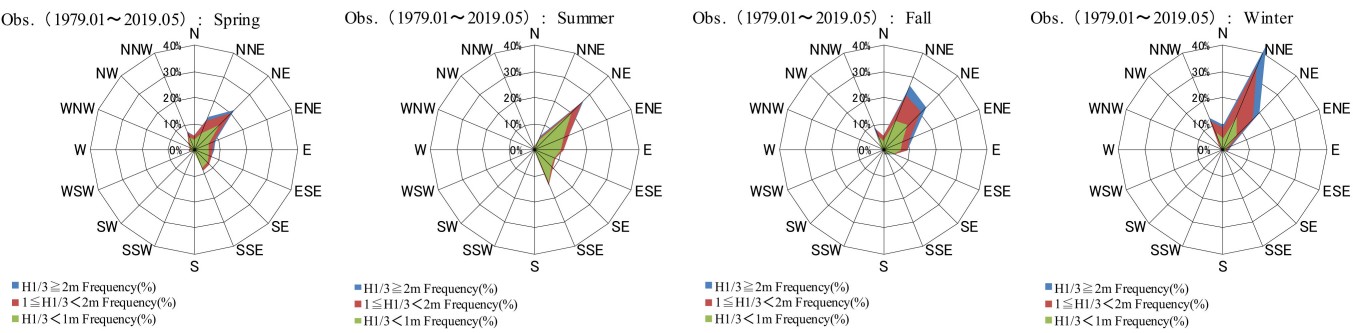

**Figure 15.** Energy-averaged wave-direction rose diagrams by season.

### 4.4. Wave Transformation

Arc-shaped sand bars—which are representative topographical nearshore features along the east coast of Korea—are distributed worldwide and have different topographical (straight and pocket beach) features and spatiotemporal scales of tidal range (van Enckevort et al., 2004) [35]. The topography of the East Coast of Korea with distinct seasonal wind changes and microtidal sea environment characteristics does not exhibit highly variable beach conditions but is characterized by the continuous maintenance of topographical changes of the

nearshore and arc-shaped sand bars due to continuous high waves (Athanasiou, 2017) [36]. Thus, Maengbang Beach clearly exhibits seasonal winds and topographical characteristics, and in particular, wave transformation occurs depending on the position of the lateral line (Mb. 1–225, 5875 m) due to the formation of crescentic bars. Therefore, wave characteristics were analyzed in this study while considering the refraction, shoaling, reflection, and diffraction caused by coastal structures using the SWAN (Simulating Waves Nearshore) wave model, which is used globally to calculate the wave transformation by a lateral line. As shown in Figure 16, the calculation domains were nested to minimize the distortion of shoaling. The grid size of domain 2 was set as 30 m to calculate the wave transformation by the lateral line.

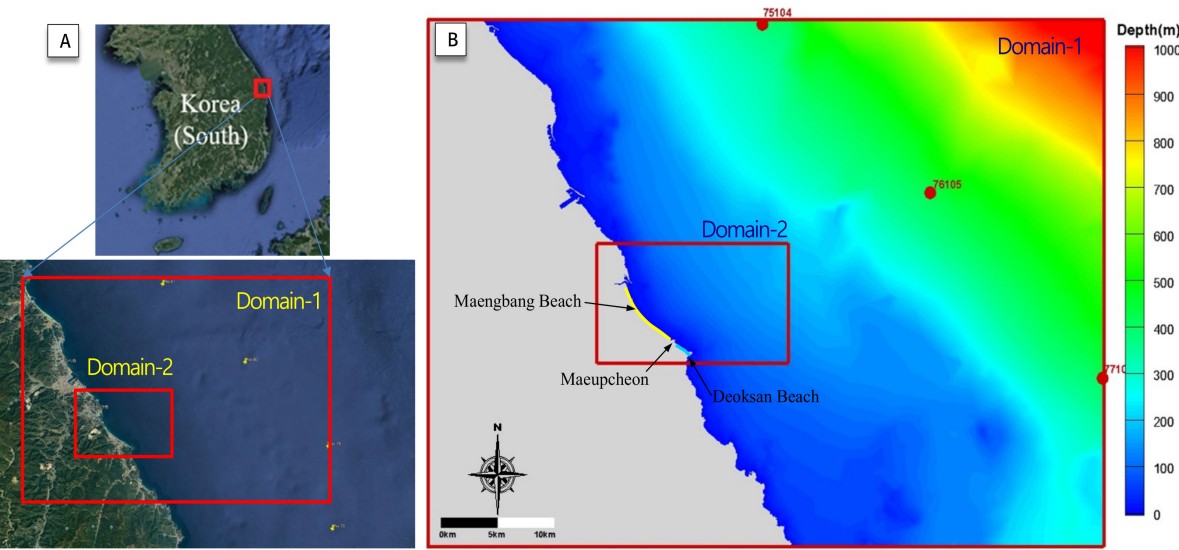

**Figure 16.** (**A**) Calculation domains of SWAN; (**B**) depth map.

### 4.5. Particle Size Distribution

To consider the seabed conditions of the target sea area, 64 nearshore and offshore points were selected, and seabed surveys were conducted six times a year for 3 years (2017–2019) to examine the characteristics of sediments around the Maengbang sea area. The average particle size distribution at each point in 2017 is presented in Figure 17. As a shoreline-change model that can consider the shape of multiple particle sizes is applied in this study, a multi-particle-size distribution was calculated instead of a single-particle-size ($D_{50}$) distribution for the sea area and effective sediment volume. For examining the particle size distribution of the mixed particle sizes representative of the sea area, the observation values over a period of three years were averaged for each position, and the particle size distribution according to the five particle groups (Figure 18B) was linearly interpolated for each line (Mb. 1–225). Fine-grained sediments of 0.15 mm or less, which do not contribute directly to shoreline change, were excluded from the study. Based on the grain size range (D = 1–5), the median grain size (D1–D5) was determined, and sediment ratios were calculated for each median grain size (Figure 18C). Based on these results, the sediment representative of the entire sea area was used as the input material for the shoreline model in the mixed grain size for each line and not in the form of median grain size (Mb. 1–225). Furthermore, because the particle size distribution of the effective sediment volume supplied from the river was not surveyed, the particle-size distribution of the effective sediment volume was established using the results of the seabed survey at the nearest point (L01) from the Maeupcheon estuary.

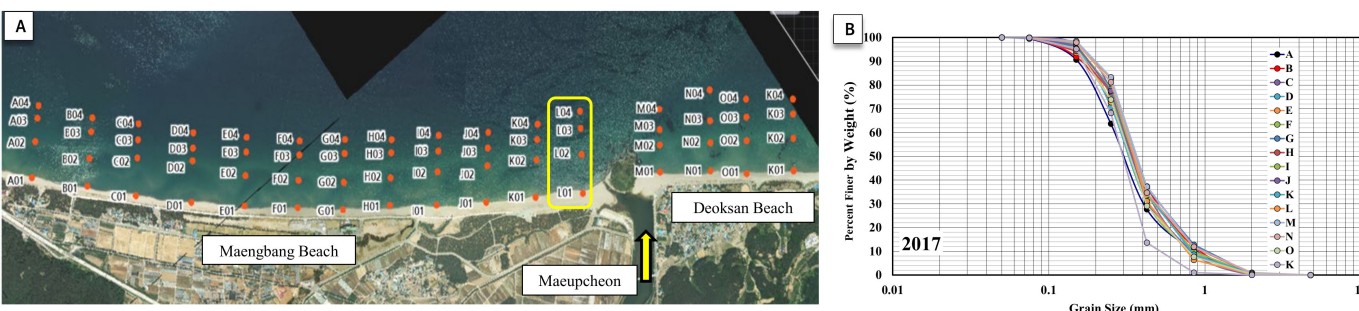

**Figure 17.** (**A**) Seabed survey locations; (**B**) seabed particle-size distribution (2017).

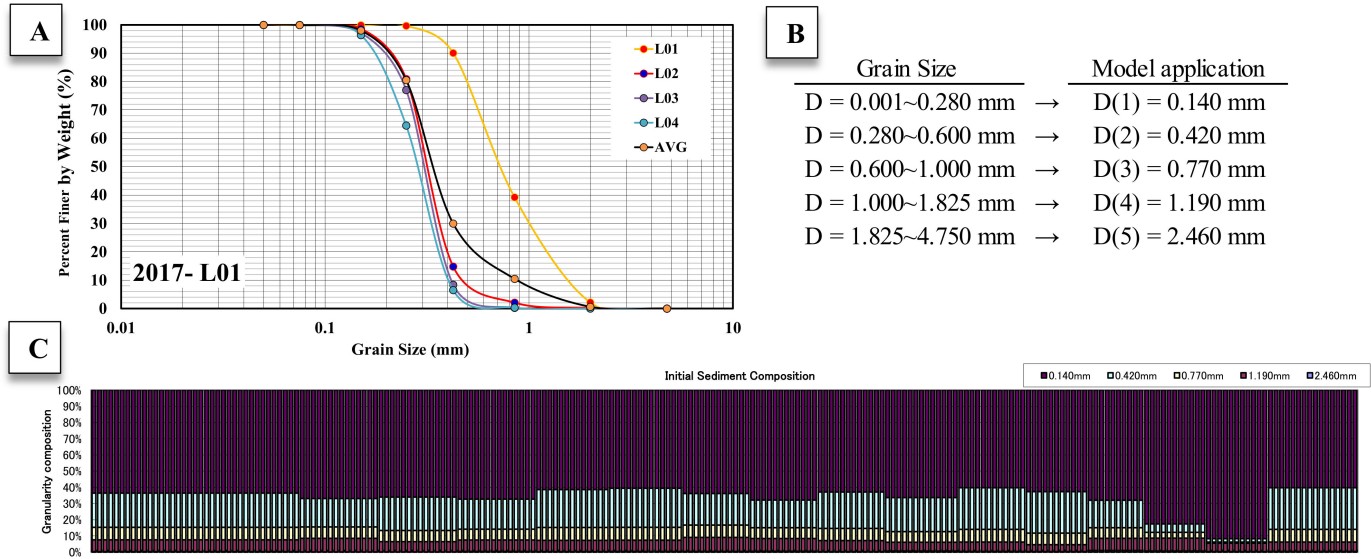

**Figure 18.** (**A**) Particle-size distribution map of the seabed at L01 point in 2017; (**B**) particle grouping by particle size; (**C**) particle-size distribution map by lateral line.

*4.6. Experimental Conditions*

The prediction performance of the shoreline model was examined to analyze the integrated sediment budget. To consider the characteristics of the study site, the effective sediment volume supplied from the river was considered. Furthermore, to take into consideration the effects of Deokbongsan (island) in front of the river mouth bar and the shapes of multiple particle sizes according to the particle-size distribution of the sediments of the sea area, the long-term shoreline change was predicted while considering the groin. The major parameters used in this model and the calculation conditions are listed in Table 2.

**Table 2.** Calculation conditions.

| Item | Case 1 | Case 2 | Case 3 |
|---|---|---|---|
| Model | Shoreline-change model considering multiple particle sizes (In-MPaS model) | | |
| Calculation domain | Shoreline length: 5875 m (225 lateral lines) | | |
| Facility conditions | Dams in mountain areas: not installed | | |
| | Submerged breakwater, groin, offshore breakwater, etc.: not installed | Groin installed (Deokbongsan) | |
| Calculation period | 1971–2020 (49 years) | | |

**Table 2.** *Cont.*

| Item | Case 1 | Case 2 | Case 3 |
|---|---|---|---|
| Incident wave conditions (energy-averaged wave) | 1980.01–1988.12: wave height Hb =1.35 m, period T = 6.32 s | | |
| | 1989.01–1996.12: wave height Hb = 1.29 m, period T = 6.32 s | | |
| | 1997.01–2012.12: wave height Hb = 1.33 m, period T = 6.16 s | | |
| | 2013.01–2020.12: wave height Hb = 1.28 m, period T = 5.90 s | | |
| Composition of sediment particle size | Single particle size (D50 = 0.650 mm), multi-particle size | | |
| | D = 0.001–0.280 mm → D(1) = 0.140 mm | | |
| | D = 0.280–0.600 mm → D(2) = 0.420 mm | | |
| | D = 0.600–1.000 mm → D(3) = 0.770 mm | | |
| | D = 1.000–1.825 mm → D(4) = 1.190 mm | | |
| | D =1.825–4.750 mm → D(5) = 2.460 mm | | |
| Sediment runoff volume of the estuary | Off | On | On |
| Coefficient of sediment transport rate | Coefficient of sediment transport rate A = 0.0002 | | |
| Depth of closer | h = −10.0 m | | |
| Width of mixing layer | Thickness of mixing layer: ΔDs = 10 cm | | |
| | The values of ΔDs/Hb in existing studies are not consistent but range from 1 to 8%: ΔDs was set as 5% of the breaking wave height Hb or 10 cm in this study. | | |
| Boundary conditions (longshore sediment transport rate) | • West side: Q = 0.0 m3/year (open boundary) | | |
| | • East side: Q = 0.0 m3/year (open boundary) | | |
| Lateral line interval of the model | • Δx = 25 m | | |

## 5. Results

The shoreline model was constructed to analyze the sediment budget of Maengbang Beach, and the parameters required for the model were set using the observed data. In particular, multi-particle size was considered for the particle-size distribution of the effective sediment volume supplied from the seabed and river, instead of a single particle size ($D_{50}$). The effective sediment volume was calculated using the result of the sediment runoff model and then applied to the model. The main results of this study are described in the following sections.

### 5.1. Effective Sediment Volume

The effective sediment volume was evaluated using the sediment runoff and SWAT models to determine the sediment transport rate to the sea area for the shoreline-change prediction, and the obtained result is presented in Figure 19. The result of the sediment runoff model was an overestimation as compared to that of the SWAT model in terms of the effective sediment volume in 2002–2007. While the SWAT model calculates the effective sediment volume by reflecting the annual watershed information, the sediment runoff model did not consider the cut soil volume by the watershed development (Korea Resources Corporation; Gangwao-do in Korea) during this period, thus resulting in a relatively high evaluation. However, the sediment runoff model is highly precise and easy to use in general and thus has high applicability. Moreover, a relational equation between the rainfall (mm/year) in the catchment and the effective sediment volume was determined (Figure 20). It is possible to set important input conditions for long-term shoreline-change prediction by evaluating the effective sediment volume flowing into the Maengbang sea area according to the rainfall.

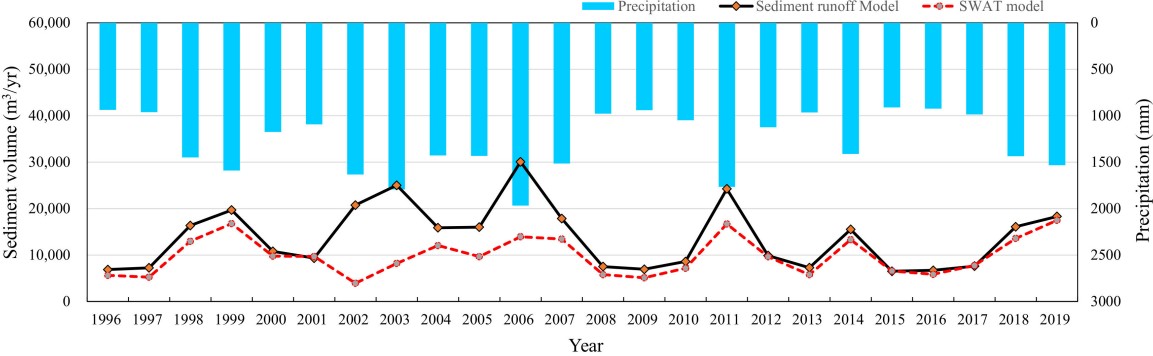

**Figure 19.** Effective sediment volume (sediment runoff and SWAT models).

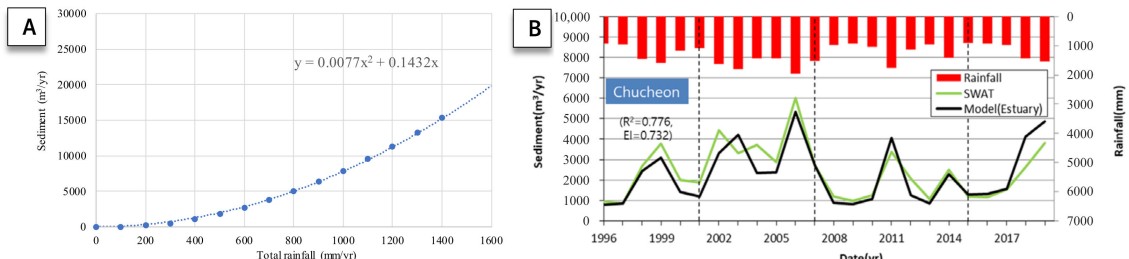

**Figure 20.** Relation between rainfall and effective sediment volume ((**A**): sediment runoff model; (**B**): SWAT models).

## 5.2. Wave Transformation

The East Sea of Korea is characterized by deep waters, a steep seabed toward the nearshore, and severe changes in incident waves. Localized wave transformation occurs when there are wide ranges of topographical features and target areas (shore length: 5.8 km) with wave transformation caused by the shallow water effect as the representative wave calculated in the study site approaches the nearshore. Therefore, the evaluation and consideration of the wave characteristics that change for each lateral line are crucial for accurately predicting the shoreline change. In this study, the wave transformation according to the incident wave condition for each lateral line acting on Maengbang Beach was calculated. The wave changes were calculated using the SWAN model, which is used globally. In addition, the changing wave height and direction at the wave breaking point by the lateral line were estimated. The results are presented in Figure 21. The wave transformation acting on each lateral line obtained from this result was used as the input data of the shoreline change model.

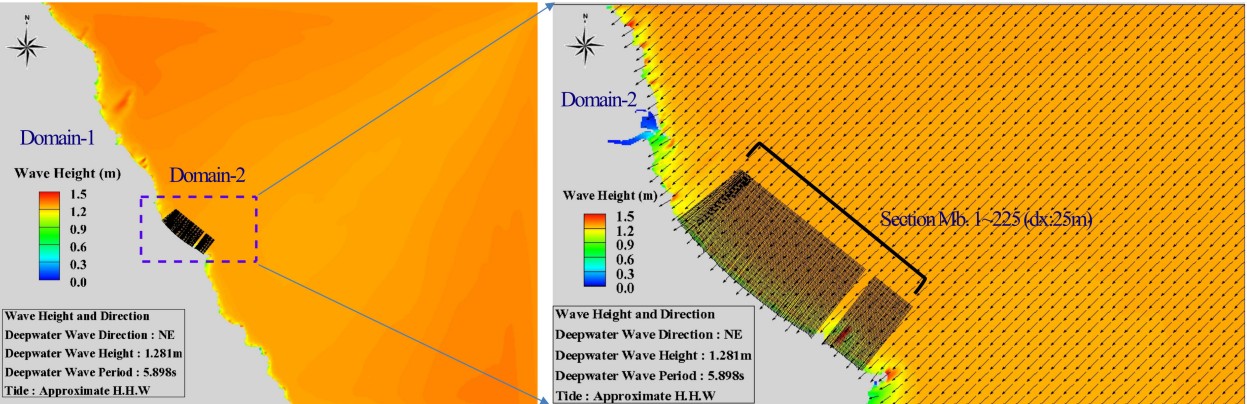

**Figure 21.** Wave transformation calculation result.

### 5.3. Shoreline Change Sensitivity Analysis

The forward and backward motion of the coastline depends on the sediment transportation patterns, and the changes in the shoreline are particularly sensitive to the characteristics of the grain size of the sediments that comprise the area and the sediments that flow into the area. The sensitivity of the change in the shoreline according to the grain size distribution and effective sediment discharge was analyzed. The results are shown in Figure 22. The results demonstrated that when the sediment supply (effective sediment discharge) to the shore was not considered, the sensitivity analysis of the change in the shoreline according to the grain size distribution (single grain size and mixed grain size) of the sediment constituting the area revealed a difference of approximately 1.3–1.9 m in shoreline change. In contrast, the change in the shoreline was more pronounced when effective sediment discharge was considered. The shoreline advanced up to 9.38 m in the single grain size distribution relative to the mixed grain size, and by almost 60% near Maeupcheon (MB. 151–225). Therefore, the characteristics of the sediment grain size distribution and the effective sediment discharge in the sea areas receiving direct sediment supply from the rivers should be considered.

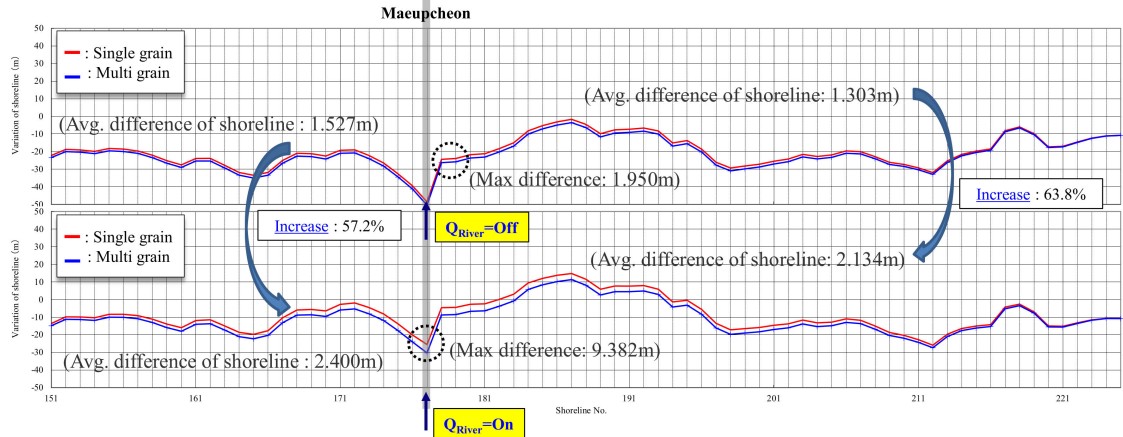

**Figure 22.** Results of shoreline change sensitivity analysis by effective sediment discharge and sediment grain size (1980–2013).

### 5.4. Shoreline Change by Particle-Size Distribution

Various models for shoreline changes in sea areas are applied for short-, mid-, and long-term prediction periods. It is important to examine longshore sediments that occur parallel to the coast owing to nearshore currents in a sea area with no river and the sediments that are transported towards nearshore and offshore areas. However, in a sea area adjacent to a river, the effective sediment volume supplied from the mountain and river to the sea is a dominant source of sediments and acts as a critical parameter in the sediment budget analysis of the sea area. Therefore, the evaluation of the effective sediment volume depending on the existence or absence of a river is crucial for predicting the shoreline change of a sea area. In this study, although there are many rivers that flow into Maengbang Beach, the effective sediment volume was not considered for rivers of which the sediments run off to the sea only during a flood (e.g., Simbang River, Figure 14). The effective sediment volume was evaluated only for Maeupcheon, which is a major river. Furthermore, a sensitivity analysis was performed based on the particle-size distribution (single particle size ($D_{50}$) and multiple particle sizes) of the sediments comprising the effective sediment volume and the sea area, and the results thus obtained are presented in Figure 23.

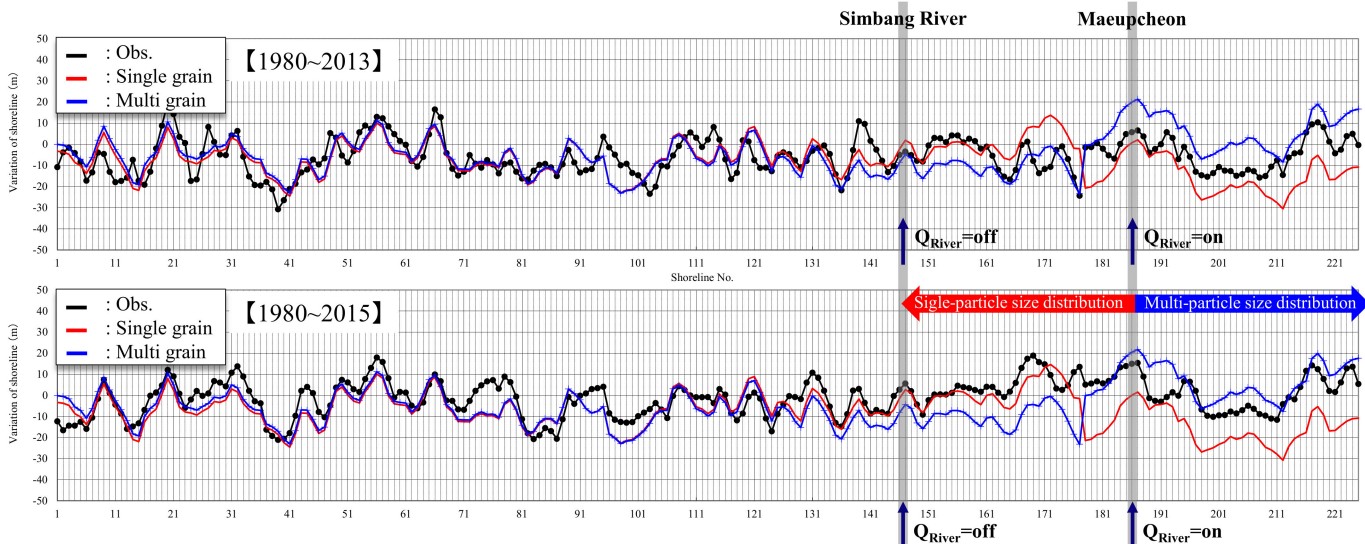

**Figure 23.** Prediction of shoreline change according to effective sediment volume (shoreline base year: 1980).

The shoreline changes in 2013 and 2015 were predicted by a calculation based on the shoreline in 1980. Based on Mb. 177, to which the effective sediment volume was supplied, the forward and backward movements occurred in the direction of the Simbang River (Mb. 135) to the left and Deoksan Beach (Mb. 225) on the right depending on the particle-size distribution of the sediments. In other areas, however, the shoreline change depending on the particle-size distribution was insignificant. This result suggests that a shoreline change occurs sensitively according to the effective sediment volume and particle-size distribution of the sediments in the sea area. In the case of Maengbang Beach, the dominant impact range of the effective sediment volume was approximately 1.0 km to the left and right.

### 5.5. Shoreline Changes Considering Topographical Features

Owing to the influence of Deokbongsan (island) in front of the mouth of Maeupcheon, the effective sediment volume supplied from Maeupcheon runs off only in the direction of Maengbang Beach. As shown in Figure 23, the precision of the prediction of the shoreline changes around the area (Mb. 175) adjacent to Maengbang Beach and Maeupcheon was unsatisfactory. This appears to be because the local characteristics of Deokbongsan, which are similar to the environment in which a groin is installed, were not considered. Hence, to take into consideration the local characteristics, the shoreline change was predicted while considering Deokbongsan in front of Maeupcheon as a groin, and the result thus obtained is presented in Figure 24. Firstly, the shoreline-change prediction result (Figure 24B) according to the supply of the effective sediment volume was compared. The result of this comparison indicates that the effective sediment volume must be considered to improve the prediction performance in an environment wherein sediments are directly supplied from a river. Furthermore, the prediction performance significantly improved when Deokbongsan was considered as a groin (Figure 24C). With respect to the distribution of sediments comprising the beach and the effective sediment volume, the prediction performance (Figure 24D) improved when a multi-particle-size distribution was considered as compared to that in the case of a single-particle-size distribution.

To evaluate the predictability of the single grain and multi grain numerical simulations, the standard errors between the observed and predicted shoreline changes (base year: 1980; predicted years: 1988, 2011, and 2013) are presented in Table 3.

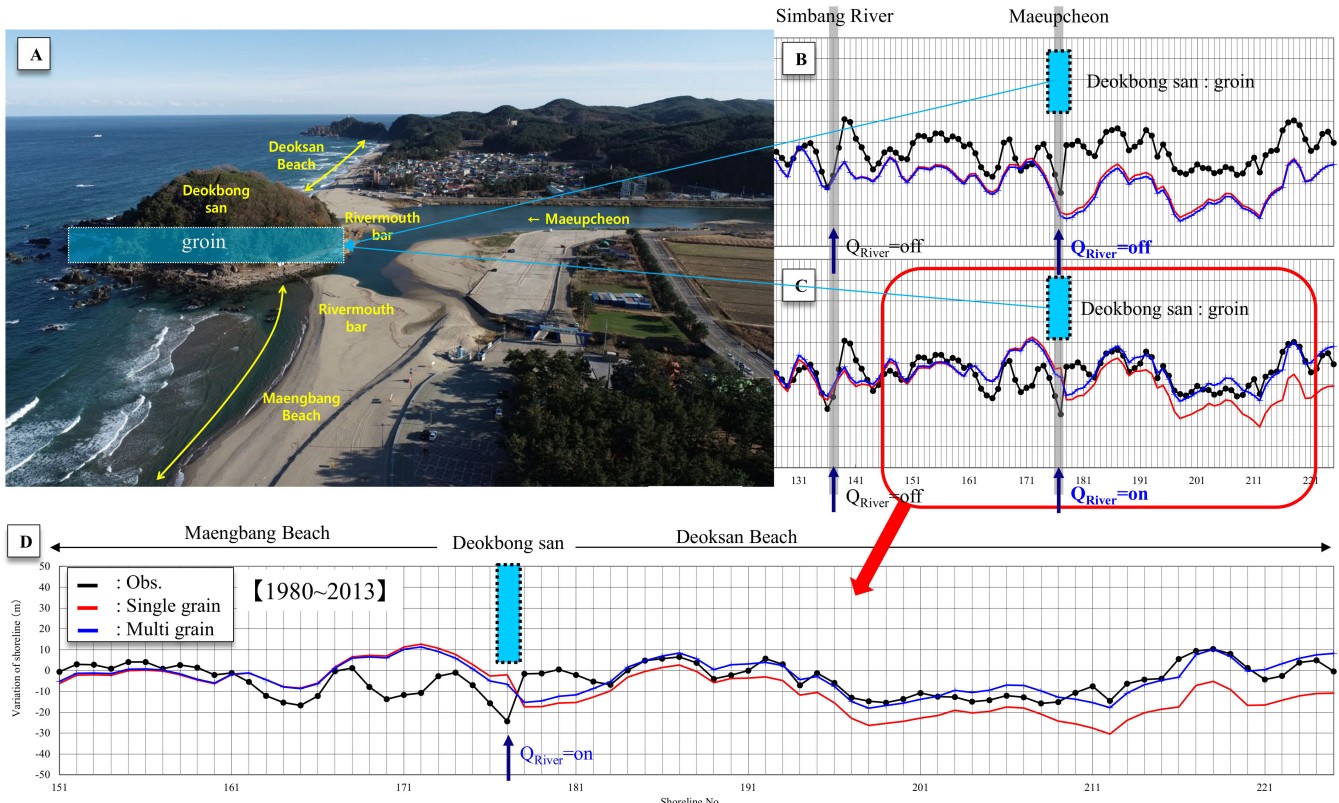

**Figure 24.** Prediction of shoreline changes while considering topographical features: (**A**) topographical features around the river mouth, (**B**) prediction result of shoreline change (effective sediment volume off), (**C**) prediction result of shoreline change (effective sediment volume on), (**D**) prediction result of shoreline change around the river mouth (effective sediment volume on).

**Table 3.** Standard error results for both particle size distributions.

| Shoreline No. | Particle Size | Single Grain | | | Multi Grain | | |
|---|---|---|---|---|---|---|---|
| | **Prediction Year** | **1988** | **2011** | **2013** | **1988** | **2011** | **2013** |
| 1–135 | | 0.43 | 0.54 | 0.61 | 0.42(▲) | 0.53(▲) | 0.61(=) |
| 135–225 | Standard error | 0.81 | 0.97 | 1.11 | 0.67(▲) | 0.65(▲) | 0.75(▲) |
| 1–225 | | 0.54 | 0.54 | 0.59 | 0.48(▲) | 0.41(▲) | 0.47(▲) |

▲—increase in the prediction rate, =—equal prediction rate.

The prediction results differed slightly according to the particle size type in the section (Mb. 1–135) with no influence from the effective sediment volume provided by the river. The prediction rate increased by at least 27% for all shoreline prediction results (1988, 2011, and 2013) that were considered mixed particle size in the section where the effect of the effective sediment volume was large (Mb. 135–225). The numerical analysis results for multigrain sizes were better in all the sections (Mb. 1–225) than single grain sizes. The results indicated that the prediction accuracy was enhanced by considering sediments in sea areas with the effective sediment volume supplied by rivers to have mixed grain sizes rather than single grain sizes.

*5.6. Calculation of Integrated Sediment Budget*

The sediment budgets of Maengbang Beach and Deoksan Beach were analyzed using the depth survey result and the prediction result of the shoreline model. Relatively recent observation data (2017–2019) were used to analyze the performance of the models, and the obtained results are presented in Figure 25. The observation data showed that the sediments

moved to the left and right from Maeupcheon (Mb. 175), which supplied the effective sediment volume. Maengbang Beach demonstrated the characteristic of a crescentic bar, in the case of which deposition and erosion occur repeatedly. The shoreline-change model (In-MPaS) result also showed a sediment transport pattern that was highly similar to the observation result. This suggests an excellent quantitative prediction performance of the sediment budget at Maengbang Beach while considering the effective sediment volume.

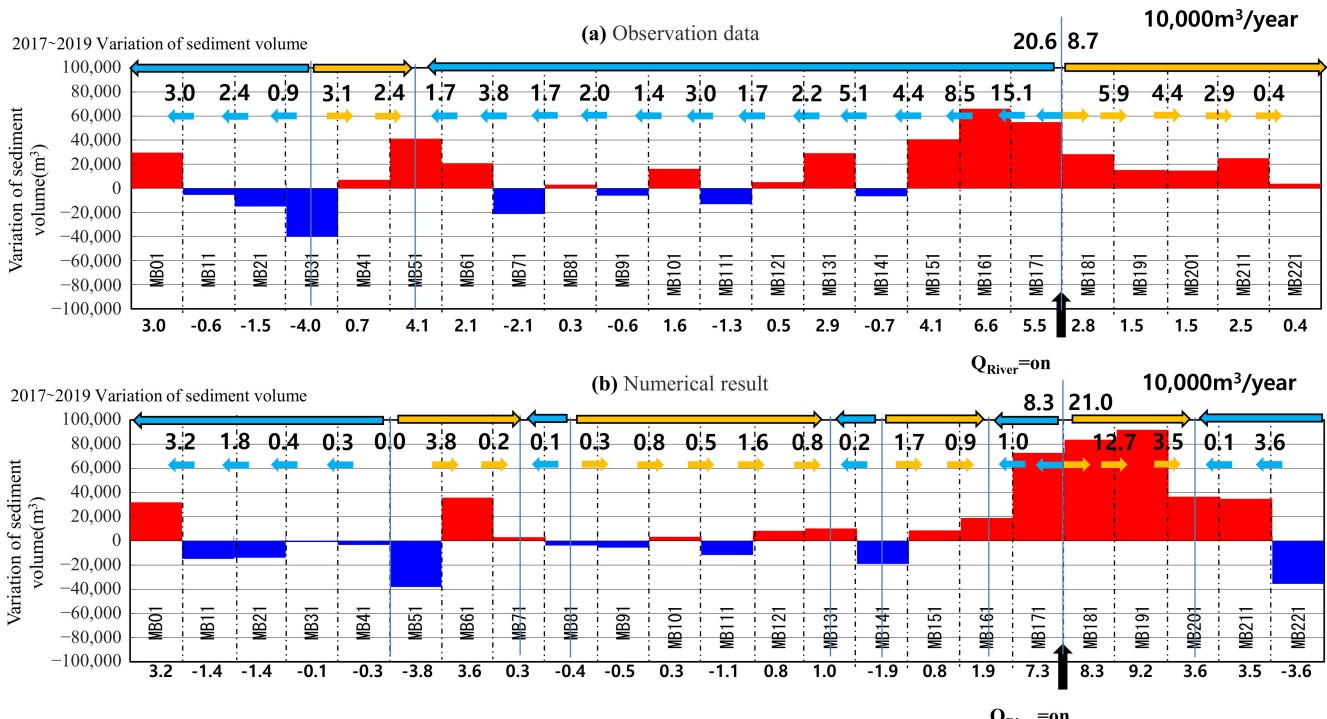

**Figure 25.** Analysis result of integrated sediment budget.

## 6. Discussion

This study presented an improved prediction performance of the shoreline model (In-MPaS) in the analysis of the sediment budget for reducing coastal erosion, and the main results of this study are as follows.

Sediment transportation on the coast can be classified into two types: parallel to the coastline (longshore sand transport) and perpendicular to the coastline (cross-shore sand transport). Longshore sand transport is often irreversible and occurs over a wide area and for an extended period, and the imbalance results in landform changes, such as coastal erosion. Cross-shore sand transport is often reversible, as witnessed when eroded beaches during high waves regain their original state during ordinary waves. However, cross-shore sand transport can result in irreversible changes to the seafloor topography or sediment loss. Therefore, it is difficult to determine the precise cause of coastal erosion. However, by analyzing the sediment budget that occurs in the ocean, the cause of the coastal erosion caused by the imbalance has been identified.

The evaluation of the long-term wave characteristics and the effective sediment volume supplied to a sea area is critical in order to predict long-term shoreline change in the sea area. The evaluation of the effective sediment volume supplied from a river is still challenging, and securing highly reliable data through observation is practically impossible. Therefore, the effective sediment volume supplied to Maengbang Beach was evaluated using the SWAT model, which has been extensively researched and verified, in order to determine the sediment runoff volume from the catchment. However, the SWAT model also has a disadvantage because it requires various observation data in the catchment in order to consider the variability of parameters required for the model. It is important to use a highly

precise model for the evaluation of the sediment volume supplied to the sea, but a model with a relatively good prediction performance that is easily applicable to areas for which no observation results are available is required. Therefore, in this study, the sediment runoff model was applied, which can be used to easily evaluate the effective sediment volume, and its prediction performance was compared with that of the SWAT model. The obtained result confirmed the high applicability of the sediment runoff model (Figure 18). The study site clearly shows the characteristics of seasonal wind and topographical features, and wave transformation occurs depending on the position of the lateral lines (Mb. 1~225, 5875 m) due to the formation of a crescentic bar. To consider such local characteristics, the incident wave transformation for each lateral line is required to be taken into consideration. In this study, the modified incident conditions for each lateral line were applied to the SWAN model for the representative wave.

The shoreline-change prediction performance was compared by the particle-size distribution of the sediments comprising the sea area and the effective sediment volume. The shoreline-change prediction model in which sediments of multiple particle sizes (five particle groups) were applied in the input condition demonstrated a better performance than the model with sediments of single particle size ($D_{50}$) as the input condition. In particular, the supply of sediments of multiple particle sizes improved the prediction performance when the effective sediment volume was considered.

To consider the topographical features of the Maengbang estuary area, Deokbongsan (island) located in front of Maeupcheon was evaluated as a groin that structurally exhibits similar performance to improve the shoreline-change prediction performance. As a result, the prediction performance of the shoreline changes on the left and right based on Mb. 175, to which the effective sediment volume was supplied, improved significantly. This result indicates the importance of establishing an appropriate model according to the topographical features. In order to upgrade the prediction of shoreline changes around the estuary (Mb. 161–181), as shown in the result of Figure 24D, additional runoff sediments due to the collapse of the river mouth bar resulting from the rapid increase in the flow of Maeupcheon from rainfall accompanied by typhoons and intensive rainfalls, as well as the sediment inflow owing to the recovery of the river mouth bar that had collapsed due to ordinary waves after a rain event, should be considered. This will improve the shoreline-change prediction performance.

## 7. Conclusions and Future Works

In this study, the characteristics of the sea area were analyzed, and model parameters were set using the observation data collected over several years to analyze the sediment budget, which can be used as important information for establishing measures to reduce coastal erosion. In addition, long-term shoreline changes were predicted by evaluating the effective sediment volume supplied from Maeupcheon to Maengbang Beach, where sediments are directly supplied to the sea area.

The evaluation of the effective sediment volume requires a quantitative model that has very large hydrological variability and can represent long-term changes. Therefore, in this study, the effective sediment volume was evaluated by applying the sediment runoff model, which makes it easy to calculate the parameters required by the model. The obtained result was similar to that of the SWAT model, which has been extensively researched and verified globally.

The prediction performance of the shoreline change of Maengbang Beach with distinct local characteristics can be improved by considering a multi-particle-size distribution instead of a single-particle-size distribution for the sediments comprising the sea area and the effective sediment volume. The shoreline-change prediction performance around the estuary was improved by considering the effective sediment volume, the transport of longshore sediments, and the effect of Deokbongsan (island) located in front of the river mouth bar as a groin, which has a structurally similar performance. The prediction performance of the shoreline model is important for realizing an accurate analysis of the

sediment budget in the sea area. To improve the prediction performance, it is important to set the representative wave for each lateral line, evaluate the effective sediment volume supplied from the river, and establish a model that can reflect the topographical features and the particle-size distribution (multi-particle size) of the sediments comprising the sea area and the effective sediment volume. In the future, we intend to improve the sediment runoff model by securing long-term observation data to verify the effective sediment volume supplied from the estuary based on this study. In addition, we intend to analyze the characteristics of the sediment transport in the sea area along with the effective sediment volume and conduct research to upgrade the shoreline prediction model while taking into consideration the sediment transport volume.

**Author Contributions:** Conceptualization, Y.-J.K. and J.-S.Y.; methodology, Y.-J.K.; software, Y.-J.K.; validation, Y.-J.K.; formal analysis, Y.-J.K.; data curation, Y.-J.K.; visualization, writing—original draft preparation, Y.-J.K.; writing—review and editing, Y.-J.K.; supervision and project administration, J.-S.Y. All authors have read and agreed to the published version of the manuscript.

**Funding:** This research was a part of the project titled "Practical Technologies for Coastal Erosion Control and Countermeasure", funded by the Ministry of Oceans and Fisheries, Korea (No. 20180404), and work was supported by a grant from the Research year of Inje University in 20210002.

**Institutional Review Board Statement:** Not applicable.

**Informed Consent Statement:** Not applicable.

**Data Availability Statement:** No new data were created or analyzed in this study. Data sharing is not applicable to this article.

**Acknowledgments:** This research was a part of the project titled "Practical Technologies for Coastal Erosion Control and Countermeasure", funded by the Ministry of Oceans and Fisheries, Korea (No. 20180404), and work was supported by a grant from the Research year of Inje Uni-versity in 20210002. The authors would like to thank the editor and four reviewers for their valuable comments and constructive suggestions on this manuscript.

**Conflicts of Interest:** The authors declare no conflict of interest.

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
