# Peer review of "Prediction of Shoreline Change for the Calculation of the Integrated Littoral Sediment Budget"

_water, doi:10.3390/w14020232_

Round 1

Reviewer 1 Report

In this manuscript, the authors take into account wave characteristics, effective sediment volume, multiple-particle sizes and geographical conditions, and improve the traditional shoreline change model, and verify the model with years of observation data. This work is of great significance for the prediction of marine sediment budget.

If the author could improve the following contents, this paper will be more readable and better met the acceptance standards:

  • The expression of the article needs to be improved. At present, there are too many long sentences, which make it difficult for readers to read and obtain effective information. For example, line 17-line 20 in the abstract.
  • The description of Figure 1 is not clear.
  • Check for errors such as Km=0.6 on line 221 and subsequent “0~0.02” in parentheses. Do the numbers in parentheses refer to the range of Km? If so, does it mean that using 0.6 in the calculation is wrong?
  • In the flow chart shown in FIG. 13, whether the first "yes" and "no" should be in the judgment freeze (the diamond). If the calculation result is "No", how to do next? Return to a previous step or end the process? I think this flow chart should be supplemented in detail.
  • Are the left and right figures in Figure 14b both profiles of MB.90?
  • Why those particle sizes are selected in the model is not clearly explained in FIG. 19b. Why are those particle sizes considered to represent multiple-particle sizes?
  • The actual observation data mentioned in the manuscript are from 2017 to 2019, but the coastlines of 2013 and 2015 are used in the verification process. And the data source is not indicated, as shown in Figure 23, "2013-1980 (OBS)".
  • How does the author consider the causes of the erosion and deposition as mentioned in the abstract (line 11-13).

Reviewer 2 Report

The subject (numerical prediction of shoreline change) is very active and interesting. Here, the sudy sound promising (with a lot of data collected). However, I find the results not presented clearly enough to be of real value.

First, I would appreciate a shorter section 2 (which is generic blabla of low value) but a more detailed section 3. Figure 13 would have been advantageously replaced by a pseudo-algorithm using mathematical notations unified with the text.

Second, the authors seem to strongly rely on the shoreline change model [28] for application to their context and they claim it is successful _in a very refined/advanced form, with many parameters_ (multi-class in particular, with time-dependent content). It would have been preferable to introduce complexities progressively (one class, fixed content) and show more closely preliminary results first (next more classes, variable content) with a basic sensitivity analysis (SA) which indicates how parameters have to be adjusted priorily.
At present, I do see SA results of that type in Fig. 23 but

  • I can hardly read the figure (too small)  
  • it seems there is no discussion of the influence of (13-14), parameters B and D_s in particular
  • what is the influence of initial conditions (for $\mu$ in particular) ? how are they determined ?
  • what is the influence of wave energy flux ?

So neither Fig 23 nor 24 show convincing results to me.

When the authors say "multiclass improves the results" it is in a very qualitative manner which I do not see from my side.

Round 2

Reviewer 1 Report

The author has revised the paper in detail according to the opinions. At present, the paper has a clear introduction of research methods, and the information in figures is relatively perfect, this work is of great significance for the prediction of marine sediment budget. However, I think line 17-20 is just an example in my first condition suggestion. The author needs to browse the full text by himself and modify the expression.

Reviewer 2 Report

The authors have answered my concerns about scientific soundness, not  about the quality of presentation. Of course, this is in part their problem and it is to the editor to decide whether the improvement is enough for publication.